

# Characterization of retinal regeneration in adult zebrafish following multiple rounds of phototoxic lesion

Alexandra H. Ranski, Ashley C. Kramer, Gregory W. Morgan[†], Jennifer L. Perez and Ryan Thummel

Department of Ophthalmology, Visual and Anatomical Sciences, Wayne State University School of Medicine, Detroit, MI, USA
[†] Deceased.

## ABSTRACT

Müller glia in the zebrafish retina respond to retinal damage by re-entering the cell cycle, which generates large numbers of retinal progenitors that ultimately replace the lost neurons. In this study we compared the regenerative outcomes of adult zebrafish exposed to one round of phototoxic treatment with adult zebrafish exposed to six consecutive rounds of phototoxic treatment. We observed that Müller glia continued to re-enter the cell cycle to produce clusters of retinal progenitors in zebrafish exposed to multiple rounds of phototoxic light. Some abnormalities were noted, however. First, we found that retinas exposed to multiple rounds of damage exhibited a greater loss of photoreceptors at 36 hours of light damage than retinas that were exposed to their first round of light damage. In addition, we found that Müller glia appeared to have an increase in the acute gliotic response in retinas exposed to multiple rounds of light treatment. This was evidenced by cellular hypertrophy, changes in GFAP cellular localization, and transient increases in *stat3* and *gfap* expression. Finally, following the sixth round of phototoxic lesion, we observed a significant increase in mis-localized HuC/D-positive amacrine and ganglion cells in the inner plexiform layer and outer retina, and a decreased number of regenerated blue cone photoreceptors. These data add to recent findings that retinal regeneration in adult zebrafish occurs concomitant with Müller glia reactivity and can result in the generation of aberrant neurons. These data are also the first to demonstrate that Müller glia appear to modify their phenotype in response to multiple rounds of phototoxic lesion, exhibiting an increase in acute gliosis while maintaining a remarkable capacity for long-term regeneration of photoreceptors.

## INTRODUCTION

Adult zebrafish (*Danio rerio*) have the ability to regenerate numerous different tissues including cardiac tissue, retinal tissue, and large portions of their appendages (i.e., fins) (*Chablais et al., 2011*; *Gonzalez-Rosa et al., 2011*; *Poss, Wilson & Keating, 2002*; *Schnabel et al., 2011*). These capabilities are contrasted in mammalian species by the formation of scars in these tissues rather than complete regeneration. Though the aforementioned tissues in the zebrafish appear to harbor the same populations of resident cell types as their

Corresponding author
Ryan Thummel,
rthummel@med.wayne.edu

mammalian counterparts, i.e., Müller glia in the retina and fibroblasts in the skin, their response to damaging stimuli is quite different. This difference in response to damage results in regeneration in one species but gliotic/fibrotic scar formation in the other. In addition, there is evidence that multiple tissues in the zebrafish rely on shared pathways for these regenerative capabilities. For example, the genes *mps1* and *hspd1,* which are necessary for proper regeneration in the heart and fin, are also upregulated in the regenerating zebrafish retina, indicating that although the details of regenerative mechanisms in each tissue might be different, there may be overlapping genetic networks at play in zebrafish tissues that allow them this unique regenerative quality (*Qin, Barthel & Raymond, 2009*).

Müller glial cells are the resident cell type in the vertebrate retina whose homeostatic function is neuronal support, including neurotransmitter recycling and production of glutamine synthetase (*Bringmann et al., 2006*) In both the zebrafish and the mammalian retina, photoreceptor ablation by intense light exposure initially results in Müller cell gliosis, including hypertrophy and upregulation of *gfap* (*Thomas et al., 2016*). However, in the zebrafish, this response is then down-regulated as Müller glia generate large numbers of retinal progenitors (*Thummel et al., 2008a*). A similar phenomenon also occurs in the zebrafish skin, as a laser-induced full-thickness wound results in initial scar formation that regresses as the epidermis is restored (*Richardson et al., 2013*). These data suggest that permissive species, such as zebrafish, use the complementary roles of a reactive state, as seen in retinal gliosis and skin fibrosis, with a subsequent progenitor proliferation to successfully restore damaged tissue.

Multiple studies have indicated that by inhibiting some of the pathways that zebrafish use to regenerate their tissues, the scar formation seen in mammals can be mimicked. For example, inhibition of cardiomyocyte proliferation by *mps1* inhibition resulted in persistent scar formation in the heart (*Poss, Wilson & Keating, 2002*). Similarly, inhibition of Fgf signaling during full-thickness wound healing also resulted in persistent scar formation in the epidermis (*Richardson et al., 2013*). Lastly, multiple rounds of amputation/regeneration of fin tissue resulted in normal regeneration of new tissue, but an increased deposition of collagen in the underlying and existing tissue (*Azevedo et al., 2011*). These studies support growing evidence that damaged tissue initially results in a gliotic/fibrotic response, which is then followed by a regenerative response. Furthermore, while many genes that control the proliferation response in these tissues are known, signals that affect gliosis/fibrosis versus stemness in permissive species like zebrafish are largely unknown. Additionally, the response of Müller glia to more than one round of phototoxic damage has not yet been described in the literature to the best of our knowledge. Therefore, the purpose of this study was to determine whether multiple rounds of light exposure would lead to a gliotic response and/or failure to regenerate.

Here we characterize, for the first time, the response of the adult zebrafish retina to multiple rounds of damaging stimuli, with a special interest in both the gliotic and stem cell responses of the resident Müller glia. We first hypothesized that upon consecutive damage to the zebrafish retina, that the Müller glia would respond randomly to the damage event, and not recruit the exact same population of Müller glia for both the first and the second light exposures. We exposed dark-adapted adult albino zebrafish to four days of intense

light to destroy the rod and cone photoreceptors and then observed a full regeneration of the retinal tissue by 28 days after light onset. Next, we exposed the same fish to a second round of light damage. Consistent with our hypothesis, we found that not all of Müller glia that re-entered the cell cycle during the first round of regeneration responded again during the second round of regeneration.

Secondly, we hypothesized that treating the retina with intense light multiple times would result in Müller glia with a more robust gliotic phenotype. When we analyzed the regenerative potential of the retina following six rounds of phototoxic treatment, we found that Müller glia still re-entered the cell cycle to produce retinal progenitors. However, we observed signs of increased gliosis among the Müller glia as these events occurred, including an upregulation of *gfap* and a redistribution of GFAP protein, cellular hypertrophy, and changes in gene expression associated with persistent gliosis. In addition, we observed significantly increased aberrant localization of inner retinal neurons following six rounds of damage and a decreased number of blue cones. Therefore, these studies illustrate that subsets of Müller glia are capable of response to multiple rounds of photolytic damage and regenerate photoreceptors. However, increasing the number of damage and response events required of this Müller glia stem cell population results in an increase in the acute gliotic response.

## MATERIALS AND METHODS

### Fish maintenance

Two fish lines were used for these studies: adult (9–18 months) *albino* (*alb*) and adult (9–18 months) Tg(*gfap:egfp*)/*alb* zebrafish. Unless otherwise noted, fish were maintained under a daily light cycle of 14 h light (250 lux):10 h dark at 28.5 °C (*Westerfield, 1995*) and fed a combination of flake food and brine shrimp. The Institutional Animal Care and Use Committee at Wayne State University approved the phototoxic lesion protocol used in this study (Protocol # 16-01-037).

### Intense light exposure

A photolytic damage model was used to destroy rod and cone photoreceptors (*Thomas et al., 2012*). First, five adult albino fish per treatment group were dark-adapted for 10 days. For the untreated fish (0 h control group), eye tissue was harvested after the 10 day dark-adaptation. For fish that were exposed to photolytic damage to photoreceptors, following the 10 day dark-adaption, animals were first exposed to approximately 100,000 lux for 30 min using a broadband light source (*Thomas et al., 2012*). Next, fish were transferred to 1.8-liter tanks and exposed to approximately 10,000 lux of light from four, 250 W halogen lamps (*Thomas et al., 2012*). Fish were subjected to constant light exposure for up to four days, and then transferred back to standard light:dark conditions. For fish that underwent 1 round of light treatment (1× treatment group), eye tissue was collected at the time points indicated. These animals were 9–10 months of age. For fish that underwent multiple rounds of light treatment (6× treatment group), following the first light treatment, fish were allowed to recover for 28 days, dark-adapted for 10 days, and then exposed to another round of light lesion as described above. This was repeated for a total of six rounds

of light treatment and 28 day recovery period. As these fish were now 18 months of age, the experiment was terminated after six rounds in order to not exceed our normal window of 9–18 months of age for light treatments.

## BrdU incorporation

For the BrdU study, adult (9–18 months) *albino* fish were light treated for 24 h. Next, fish were transferred to a 1L solution containing 0.66 g of NaCl, 0.1 g Neutral Regulator, and 1.5 g BrdU (SeaChem Laboratories, Inc. Stone Mountain, GA, USA; Sigma). After 24 h, these fish were transferred to normal light:dark conditions for 28 days, dark-adapted for 10 days, and then exposed to another round of light lesion as described above. The timeframe for BrdU incubation was based on previous studies which indicated that all participating Müller glia re-enter the cell cycle within this window, starting at ∼31 hpL, and that the first daughter cell is visible at 48 hpL (*Kassen et al., 2008*; *Thummel et al., 2008a*; *Vihtelic & Hyde, 2000*). Thirty-six hours after starting the second treatment, fish were euthanized and whole eyes were enucleated and processed for immunohistochemistry. The percentage of Müller glia in each group (BrdU +, PCNA +, double-positive) were calculated within a linear distance of 300 microns. The Kruskal-Wallis nonparametric H test was used to assess whether there were significant differences between the groups at the 0.05 significance level.

## Immunohistochemistry and confocal microscopy

Immunohistochemistry was performed as previously described (*Thummel et al., 2008a*). Briefly, five whole eyes from each experimental group were harvested from sacrificed adult (9–18 months) animals and were fixed in 9:1 ethanolic formaldehyde (100% ethanol: 37% formaldehyde) overnight at 4 °C. Eyes were cryopreserved, embedded in freezing medium, and retinal sections (16 $\mu$M) containing or immediately surrounding the optic nerve were collected on Superfrost Plus glass slides (Fisher Scientific, Pittsburgh, PA, USA). Two different immunohistochemistry protocols were used for the primary antibody incubation. Standard immunohistochemistry was performed as previously described (*Thummel et al., 2008a*) using the following primary antibodies: mouse Zpr-1 antibody (Zebrafish International Resource Center, Eugene, OR, USA; 1:200), rabbit anti-Rhodopsin antisera (gift from David Hyde; 1:5,000), rabbit anti-GFP (Abcam; 1:1,500), mouse anti-PCNA (Sigma; 1:1,000), rabbit anti-Blue opsin (gift from David Hyde; 1:500), rabbit anti-UV opsin (gift from David Hyde; 1:1,000), mouse anti-Glutamine Synthetase (Chemicon; 1:500), and mouse 4C4 (gift from Peter Hitchcock; 1:250). For antibodies requiring antigen retrieval, the following procedure was used. Slides were preheated to 55 °C on a slide warmer while a glass coplin jar was preheated in an Oster 6-quart Digital Food Steamer for 30 min . Sodium Citrate Buffer (10 mM Sodium Citrate/1XPBS/0.05% Tween-20, pH 6.0), heated in a microwave until boiling, was poured into the coplin jar and the slides were immediately placed in the buffer solution. After 30 min, the coplin jar was removed from the steamer and allowed to cool at room temperature for 30 min. The slides were removed and the tissue was outlined with a Pap Pen (Daido Sangyo, Tokyo, Japan). Slides were rinsed three times for 5 min with 1XPBS/.5% Triton X-100. Retinal sections were covered with a blocking solution of 1XPBS/.5% Triton X-100/20% sheep serum for 1 h. Sections were incubated

**Table 1  Primer sequences used for RT-qPCR.** All primers were designed to span introns using the NCBI/Primer-BLAST tool with a target product size of 115–150 base pairs.

| Target gene | Forward primer sequence (5′–3′) | Reverse Primer Sequence (5′–3′) |
| --- | --- | --- |
| cdk1 | 5′- GGT CTG GCC AGA TGT TGA GTC TC -3′ | 5′- TCG TGC CGA AAT CCT CTT AGG G -3′ |
| c-jun | 5′- AGA GAC GCA GAG CGC ATG A -3′ | 5′- CGC GTC CCT GTT TTA CTC CTA -3′ |
| pax6a | 5′- TCG AGG TTC CCT TGT TGG AC -3′ | 5′- GAC GCA CGG TTC CAA GTT TC -3′ |
| pax6b | 5′- CCA CAC CGT ACG GGA TTC AA -3′ | 5′- TCC CAG CGT CCC TCT TAT CT -3′ |
| pcna | 5′- TAC TCA GTG TCT GCT GTG GTT CC -3′ | 5′- CAT TTA ATA AGT GCG CCC GC -3′ |
| shha | 5′- AGT CTT ACC TTT CGC ATC CCC -3′ | 5′- GAT GTC CTT GCC GTC TCC TC -3′ |
| six3b | 5′- GAT AGC AGC GCA AAC ACG AC -3′ | 5′- CGC GAA ATT GGG CAG GAA AA -3′ |
| tgif1 | 5′- CCC ATC TAT CCA CAC TAC AGG TTT -3′ | 5′- CTG AGC ATT TCG CCA CCC TT -3′ |
| gfap | 5′- GCA GAC AGG TGG ATG GAC TCA -3′ | 5′- GGC CAA GTT GTC TCT CTC GAT C -3′ |
| vim | 5′- TAA GCC TGC GAG AGT CCA TGA -3′ | 5′- TCG TTT GGG TGG ACT CGT TT -3′ |
| glulB | 5′- GCC CGC TTC CTC CTA CAC A -3′ | 5′- CTC CTC AAT ATG CTT CAA ACC TCC -3′ |
| kcnj10a | 5′- CCT GTT TTC GGC CAC CTT TG -3′ | 5′- CGG CGT ATG GTT GGA TGG AG -3′ |
| rlbpa | 5′- CTG CGT GCC TAC TGT GTA ATC C -3′ | 5′- GCT CGG TGT GTT TGA TTC CAG -3′ |
| rlbpB | 5′- TGA GAC GGA TGA GAA GCG AA -3′ | 5′- ACC TCA CAA GCA CGC CAT C -3′ |
| stat3 | 5′- GAG GAG GCG TTT GGC AAA -3′ | 5′- TGT GTC AGG GAA CTC AGT GTC TG -3′ |

overnight at 4 °C in primary antibody diluted into 1XPBS/.5% Triton X-100/2% sheep serum. The primary antibodies and antisera requiring use of this protocol were: mouse anti-HuC/D (Invitrogen; 1:50) and rabbit anti-GFAP (DakoCytomation; 1:500).

For both primary antibody procedures, after overnight incubation, sections were rinsed three times for 10 min with PBS/.5% Triton X-100 at room temperature. Sections were then incubated in the secondary antibody solution diluted in 1XPBS/.5% Triton X-100/2% sheep serum for 1 h. After secondary antibody incubation, sections were rinsed in 1XPBS/.5% Triton X-100 3 times for 10 min. Slides were covered with a coverslip using ProLong Gold (Molecular Probes, Eugene, OR). Secondary antibodies included AlexaFluor-conjugated 488, 594 and 647 goat anti-primary (1:500, Life Technologies, Grand Island, NY, USA). Nuclei were stained using TO-PRO-3 (TP3; 1:750; Life Technologies, Grand Island, NY, USA) or DAPI.

Confocal microscopy was performed with a Leica TCS SP8 confocal microscope. Z-stacked images were taken within a 4 μM thickness with 0.5 μM between slices. Quantification of retinal cell number was performed by manual count of a 300 micron linear distance on the central dorsal retina using five captured images in each group of biological replicates. The differences between groups were analyzed by a one-way ANOVA if three groups were compared as seen in all photoreceptor quantifications and the quantification of mis-localized HuC/D+ neurons. A Student's $T$-test was performed using $p < 0.05$ as a statistical cut-off if only two groups were compared as in the comparison of PCNA+ Müller glia at 36 h post light (hpL) and qRT-PCR results.

### Quantitative real-time PCR

Retinal tissue was isolated from control and experimental groups at 36 hpL, 72 hpL, and 28 days post light (dpL). Pools of five retinas were collected in biological triplicate.

Tissue was collected into 1 mL of Trizol, manually homogenized, and frozen in Trizol at −80 °C. RNA was isolated using the Trizol Reagent, per manufacturer's instructions (Invitrogen). DNase treatment was not performed. RNA was quantified and assessed for purity using a nanodrop. 1 μg RNA was used as the input for cDNA synthesis, which was performed using the manufacturer's protocol with random oligos and Superscript II polymerase (Invitrogen, 18064). cDNA was diluted 1:25 in nuclease free $H_2O$ and 2 μL was used in the following reaction. Quantitative Real-Time PCR (qPCR) was carried out in technical and biological triplicate in a 10 μL reaction using the SYBR green reagent (#4309155; Applied Biosystems, Foster City, CA, USA) on a CFX Connect Real-Time System (Bio-Rad). Cycling conditions were as follows: step 1. 95 °C hold for 10 min, step 2. 95 °C denature for 15 s, 60–63 ° C anneal/extend for 1 min (cycle step 2. 39 times), step 3. 65–95 °C melt-curve increasing by 0.5 °C every 5 s. Primers were designed to span introns with the exception of the *gpia* primer pair (BIO-RAD Assay ID:qDreCED0007433) which was used as the endogenous reference gene. Primer sequences are displayed in Table 1. PCR products were not sequenced to confirm identity; however, the melt curve was analyzed to determine single peaks for each primer set. Quantification of fold-change in expression was performed between control (1×) experimental (6×) groups using the Livak $2^{-\Delta\Delta C(t)}$ method with a $p < 0.05$ for statistical cut-off (*Pfaffl, 2001*). Briefly, the mean CT for each gene was normalized against the *gpia* mean CT endogenous reference to obtain a ΔCT value. The ΔCT value of the 6× group was averaged across the biological triplicate and then normalized against the 1× group, giving the ΔΔCT value. Statistical differences between groups were determined using a student's *T*-test from the normalized ΔΔCT values. Finally, ΔΔCT values were converted to log 2-fold changes in gene expression and graphed.

## RESULTS

### Evidence that the exact same population of Müller glia do not re-enter the cell cycle following each light damage event

The timeline of light-induced damage and regeneration of the zebrafish retina has previously been described (*Kassen et al., 2007*; *Lenkowski & Raymond, 2014*; *Thomas et al., 2012*; *Thummel et al., 2008a*; *Vihtelic & Hyde, 2000*). Briefly, photoreceptor apoptosis peaks at 24 hpL, followed by Müller glia cell cycle re-entry at 31–36 hpL. The resultant progenitors continue to proliferate over the next 3 days as they migrate from the inner nuclear layer (INL) to the outer nuclear layer (ONL), where they begin to differentiate into new photoreceptors during the following week. Previous reports showed that ∼50% of the total population of Müller glia re-enter the cell cycle during the photolesion paradigm (*Thummel et al., 2008b*). Given that this study used multiple rounds of phototoxic treatment, we were first interested in testing whether the exact same subset of Müller glia that acted as stem cells following one round of damage would respond in the same way a second time. To examine this, adult *albino* zebrafish were subjected to two rounds of light damage. During the first round, Müller glia that re-entered the cell cycle were permanently marked with BrdU incorporation during the S phase of the cell cycle. During the second round of Müller glia cell re-entry at 36 hpL, retinal sections were triple immunolabeled with

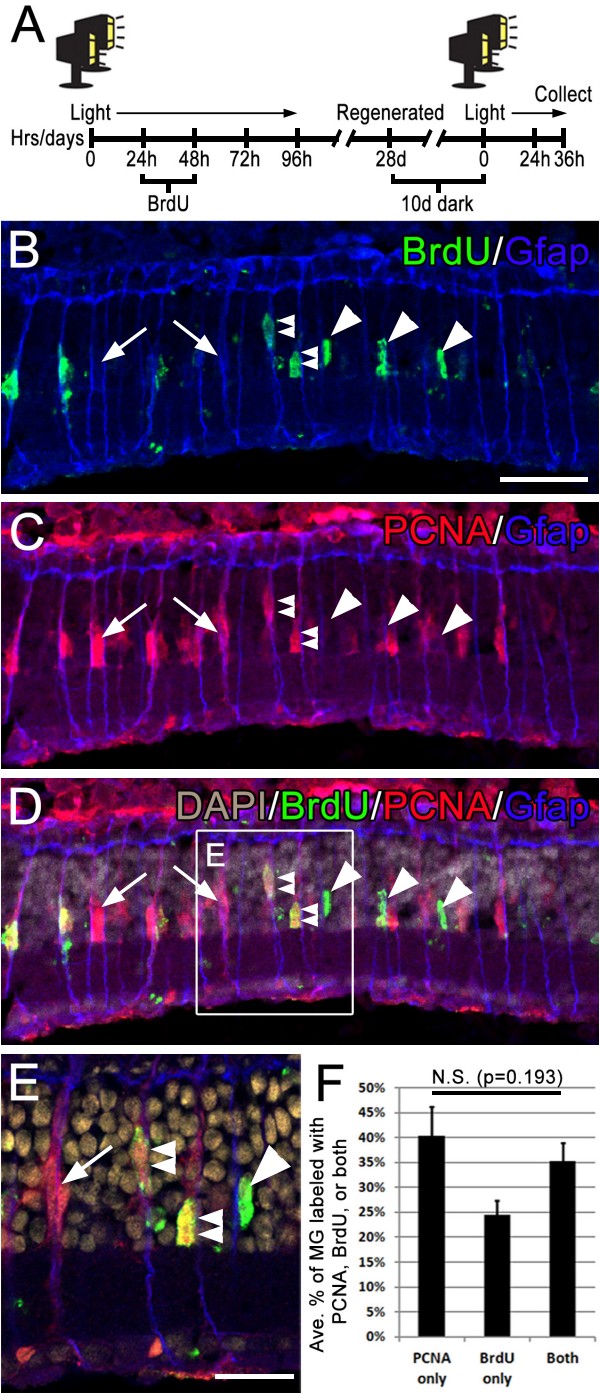

**Figure 1** **Following consecutive rounds of light damage, different populations of Müller glia re-enter the cell cycle.** (A) Timeline of the BrdU and light lesion, fish were exposed to two rounds of light lesion separated by 28 day period of standard light/dark cycling. (B–D) Retinal sections collected at 36 hpL immunolabeled with anti-GFAP (blue), anti-BrdU (green), anti-PCNA (red) and DAPI (nuclear stain; grey). Arrowheads point to BrdU-positive cells, arrows point to PCNA-positive cells, and double arrowheads point to cells co-immunolabeled with both PCNA and BrdU. (E) An inset of image D. (F) Quantification of the average numbers of BrdU-positive cells (24%), PCNA positive cells (40%) and co-labeled cells (35%) counted over a linear distance of 300 μm on the central dorsal retina following two consecutive rounds of light damage. Scale bar represents 25 μm in B and 12 μm in E.

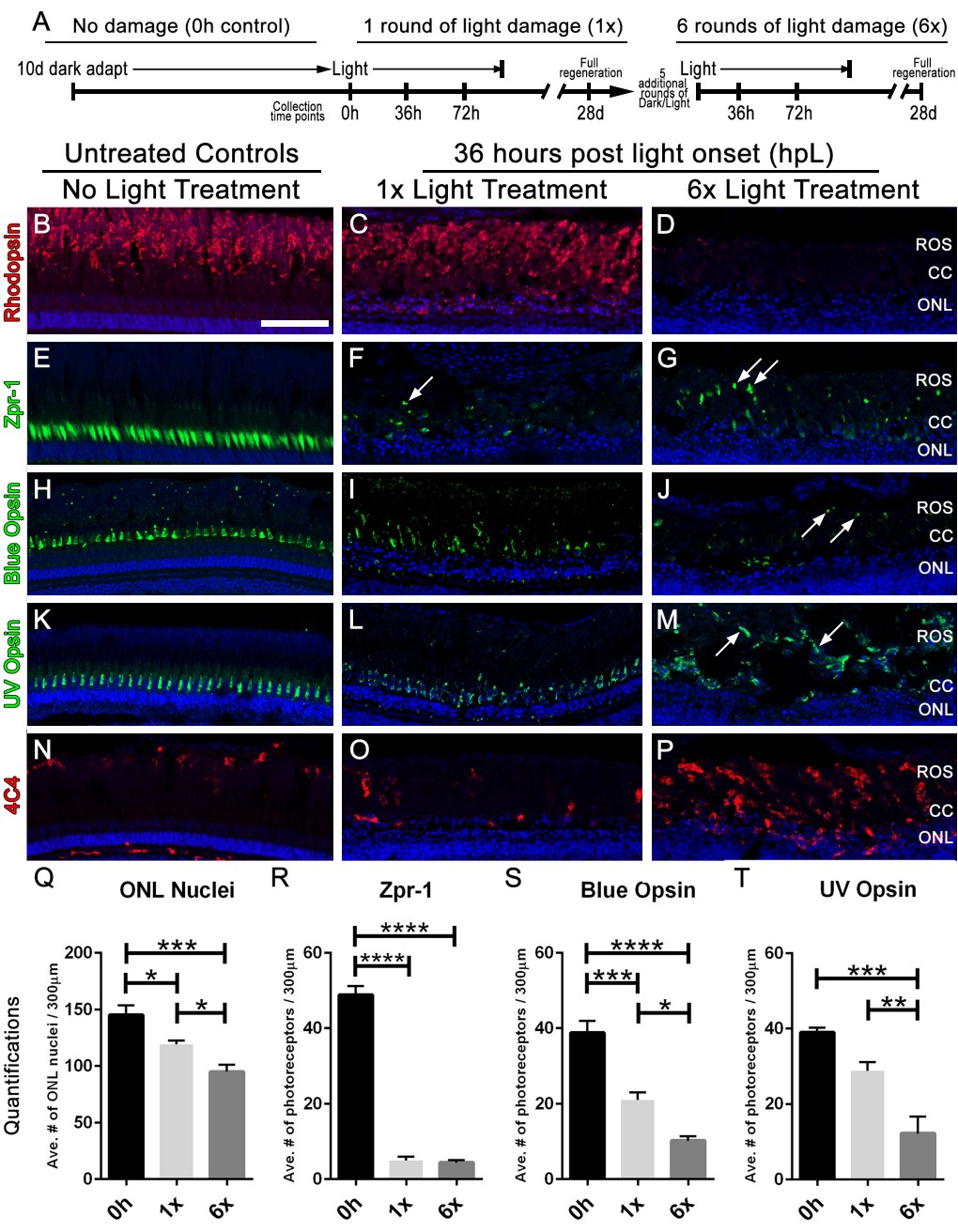

**Figure 2 Multiple rounds of light treatment leads to greater photoreceptor loss at 36 h of light treatment.** (A) Experimental design used for multiple rounds of light treatment. (B–P) Retinal sections collected at 36 hpL immunolabeled with anti-rhodopsin, Zpr-1, anti-blue opsin, anti-UV opsin, and 4C4 to show the varying amounts of damage to photoreceptors in untreated control (No Light Treatment), experimental control (1×Light Treatment), and experimental retinas (1×Light Treatment). Nuclei are stained blue with TO-PRO-3. (B–D) Rod photoreceptor outer segments are immunolabeled with anti-rhodopsin (red). (E–G) Red-green double cones are immunolabeled with Zpr-1 (green). (H–J) Long single cones are immunolabeled with anti-blue 

**Figure 2 (…continued)**
opsin (green). (K–M) Short single cones are immunolabeled with anti-UV opsin (green). (N–P) Microglia/macrophages are immunolabeled with 4C4 (red). In the images taken after six rounds of light treatment, arrows point to individual cone opsins in the large debris field. (Q–T) Quantification of average numbers of photoreceptors in untreated (0h), control (1×) and experimental (1×) groups ($n = 5$ per group) counted over a linear distance of 300 μm on the central dorsal retina. Asterisks indicates significant differences between groups (four asterisks $p < 0.0001$; three asterisks $p < 0.0003$; two asterisks $p < 0.006$; one asterisk $p < 0.02$). Scale bar represents 25 μm.

antibodies against BrdU (S phase during the first round), PCNA (G1 → S phase during the second round), and GFAP (all Müller glia). We observed that 24% of GFAP-positive Müller glia were only immunolabeled with anti-BrdU (Fig. 1, arrowheads), marking cells that re-entered the cell cycle following the first round of damage, but not the second. Conversely, 40% of GFAP-positive Müller glia were only immunolabeled with anti-PCNA (Fig. 1, arrows), indicating cells that initiated cell cycle re-entry only during the second round of light damage, but not the first. Finally, 35% of GFAP-positive Müller glia were co-immunolabeled with anti-BrdU, and anti-PCNA (Fig. 1, double arrowheads), marking cells that re-entered the cell cycle during both rounds of regeneration. A Kruskal-Wallis nonparametric H-test determined that there were not significant differences between these groups ($p = 0.101$). These data suggest that the exact same population of Müller glia do not re-enter the cell cycle following each individual light damage event.

### Multiple rounds of light damage leads to a greater loss of photoreceptors and increased Müller cell gliosis, but a normal Müller glial and progenitor cell proliferation

Next, we compared photoreceptor numbers and proliferation in untreated control animals with animals exposed to one round (1×) and six rounds (6×) of light treatment at two intermediate time points, 36 and 72 hpL (*Thummel et al., 2008a*). At 36 hpL, a one-way ANOVA analysis was performed among all photoreceptors quantified and revealed that there were statistically significant differences among the groups ($p < 0.0001$). Notably, a post-hoc test also revealed statistical differences between the 1× and 6× groups. Specifically, 6× light treated retinas exhibited decreased Rhodopsin immunolocalization (Figs. 2B–2D), rod photoreceptor nuclei in the outer nuclear layer (Fig. 2Q; $p < 0.05$) and significantly fewer numbers of long-single (blue) cones (Figs. 2H–2J, S; $p < 0.02$) and short-single (UV) cones (Figs. 2K–2M, T; $p < 0.006$) when compared with the 1× treated retinas. Zpr-1 positive double cones were nearly completely destroyed in both groups (Figs. 2E–2G, 2R; $p > 0.05$). Additionally, 6× retinas showed a significantly higher number of 4C4-positive microglia/macrophages compared with 1× retinas (Figs. 2O–2P; Average number of 13.8 vs. 46.2 per 300 μm, respectively; $p < 0.01$), which is consistent with the visually more pronounced cellular debris observed in the 6× retinas (Figs. 2G, 2J, 2M; arrows). These data suggest that retinas that have undergone 6× rounds of damage and regeneration exhibit increased photoreceptor loss to a subsequent light insult and exhibited an increased inflammatory response.

Increased photoreceptor loss would predict a higher percentage of Müller glia to re-enter the cell cycle at 36 hpL (*Thomas et al., 2012*). Contrary to this prediction, we observed that

the percentage of Müller cells that re-entered the cell cycle was not significantly different between the 1× and 6× groups (Figs. 3B–3C, 3J). In addition, at 72 hpL, both 1× and 6× retinas show Müller glia surrounded by clusters of PCNA-positive progenitors migrating to the outer nuclear layer (ONL; Figs. 4E–4F), suggesting that progenitor amplification and migration was not severely altered in 6× retinas. However, retinas from the 6× groups did show signs of increased Müller cell gliosis, including unorganized, hypertrophied Müller glia with a significant upregulation of *glial fibrillary acidic protein* (*gfap*) mRNA expression (Fig. 3K), and thickened GFAP-positive appendages in the photoreceptor layer at 36 hpL (Figs. 3E–3F, 3H–3I). Expression of *signal transducer and activator of transcription 3* (*stat3*), one of the earliest molecular markers of retinal stress, was also significantly upregulated (Fig. 3K) (*Gorsuch & Hyde, 2014*; *Nicolas et al., 2013*). At 72 hpL, retinas from the 6× groups showed continued signs of Müller cell disorganization. GFAP immunohistochemistry revealed obvious disorganization of GFAP within the Müller glia with evidence of redistribution of the GFAP fibers away from the endfeet and towards the cell body, perhaps indicative of a hypertrophy phenotype (Figs. 4A–4C). These data suggest that 6× rounds of light damage resulted in increased signs of Müller glia stress and gliosis, but did not affect their ability to re-enter the cell cycle and produce retinal progenitors.

Next, we analyzed the expression of genes that have been associated with proliferation in other cellular events, such as *cyclin dependent kinase 1* (*cdk1*), and *c-jun* (*Malumbres, 2014*; *Munshi & Ramesh, 2013*), and genes shown to be required for amplification of the progenitor population during retinal regeneration, such as *proliferating cell nuclear antigen* (*pcna*), *paired box protein 6* (*pax6a* and *pax6b*), *sonic hedgehog* (*shha* and *shhb*), *sinus oculis homeobox 3* (*six3b*), and *transforming growth factor-beta-induced factor 1* (*tgif1*) (*Fuccillo, Joyner & Fishell, 2006*; *Hatton et al., 2006*; *Kaur et al., 2018*; *Lenkowski et al., 2013*; *Sherpa et al., 2014*; *Thomas et al., 2018*; *Thummel et al., 2010*; *Thummel et al., 2008b*; *Todd & Fischer, 2015*). Compared with 1× control retinas, retinas that underwent 6× rounds of light damage exhibited a significant downregulation of many pro-proliferative genes, including *cdk1*, *pcna*, *shhb*, *six3b*, and *tgif1* (Fig. 4J). However, no change was observed for *c-jun* and *pax6a*, and one pro-proliferative gene, *shha*, was significantly upregulated (Fig. 4J). In addition, we analyzed a group of genes associated with Müller glial function of retinal homeostasis and support, such as *glutamine synthetase* (*glulb*), *potassium inward-rectifying channel* (*kcnj10a*), and *retinaldehyde binding protein 1* (*rlbp1b*) (*Bringmann & Wiedemann, 2012*). Compared with 1× control retinas, 6× retinas exhibited a significant downregulation of *glulB* and *rlbp1b* (Fig. 4J), suggesting that the Müller glia have begun to lose their ability to recycle extracellular glutamate and participate in the regeneration of the cone visual pigment. Notably, expression of *rlbp1a*, which is expressed in the retinal pigmented epithelium but not in Müller glia (*Collery et al., 2008*), was not significantly different between groups (Fig. 4J). Finally, we analyzed the expression of two intermediate filaments of the Müller glia, *gfap* and *vimentin* (*vim*), and found no differences in *gfap* expression and a significant downregulation of *vim* expression between groups at 72 hpL (Fig. 4J). Together, these data suggested a complex set of consequences to 6× rounds of light treatment in regards to maintaining proliferation of the progenitor pool and Müller glia function.

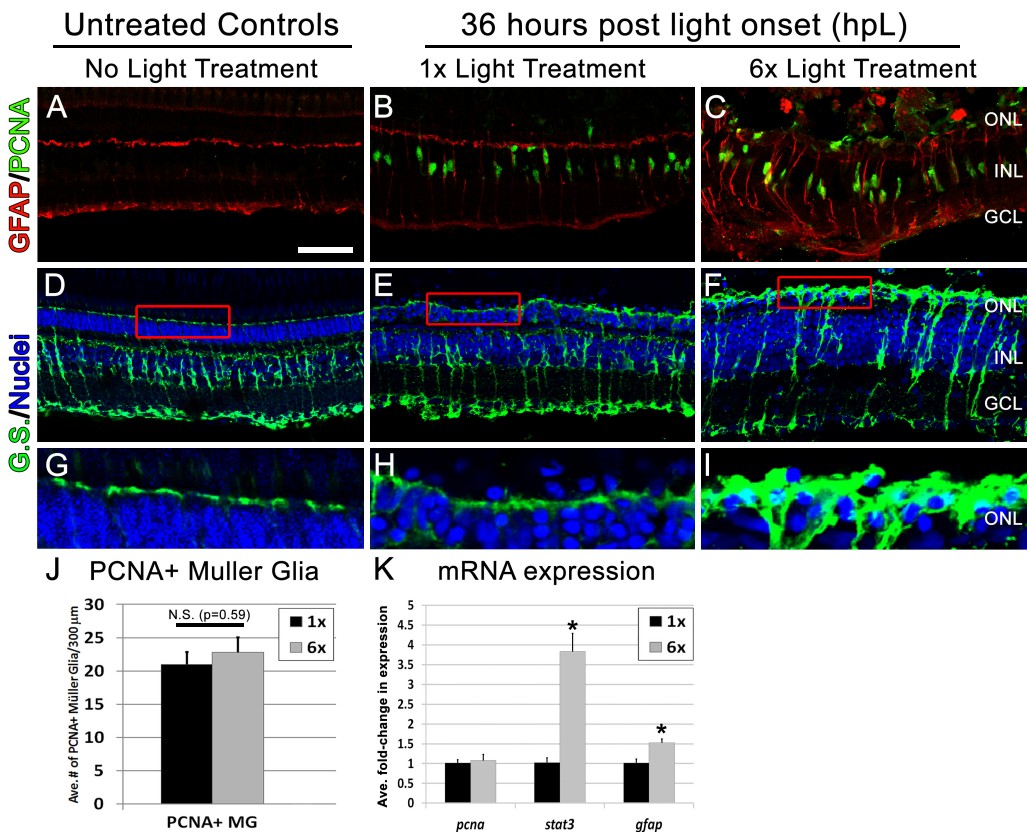

**Figure 3** **Following six rounds of light treatment Müller glia become highly unorganized and exhibit gliotic characteristics.** (A–C) Retinal sections collected at 36 hpL immunolabeled with anti-GFAP (red), and anti-PCNA (green) in untreated control (No Light Treatment), experimental control (1×Light Treatment), and experimental retinas (1×Light Treatment). (D–F) Retinal sections collected at 36 hpL immunolabeled with anti-glutamine synthetase (G.S.; green) and TO-PRO-3 as nuclear stain (blue). (G–I) Magnified inset of regions indicated by red box in (D–F). (J) Quantification of average number of PCNA positive Müller glia in experimental control and experimental groups ($n = 5$). Cells were counted over a linear distance of 300 μm on the central dorsal retina. (K) Graph showing the average fold-change in expression of genes associated with proliferation (*pcna*) and gliosis/stress (*stat3*, *gfap*) as determined by real-time qPCR. Asterisk indicates significantly different from control ($p < 0.05$). Scale bar represents 25 μm.

## Multiple rounds of light damage results in aberrant long-term regeneration of the inner retina, a reduction of regenerated long single cones, and full regeneration of all other photoreceptor subtypes

Finally, we analyzed 1× and 6× groups at 28 dpL, an established time-point for phenotypically defined full regeneration as previously described (*Thomas et al., 2016*; *Thummel et al., 2008a*; *Thummel et al., 2008b*). Immunolocalization of GFAP showed an apparent resolution in Müller cell gliosis and organization at 28 dpL in both 1× and 6× retinas (Figs. 5A–5C). Interestingly, immunolocalization of HuC/D, a marker of amacrine and ganglion cells, demonstrated aberrant localization of HuC/D-positive cells in both the 1× and 6× retinas compared with non-light treated control retinas (Figs. 5D–5F).
_______________

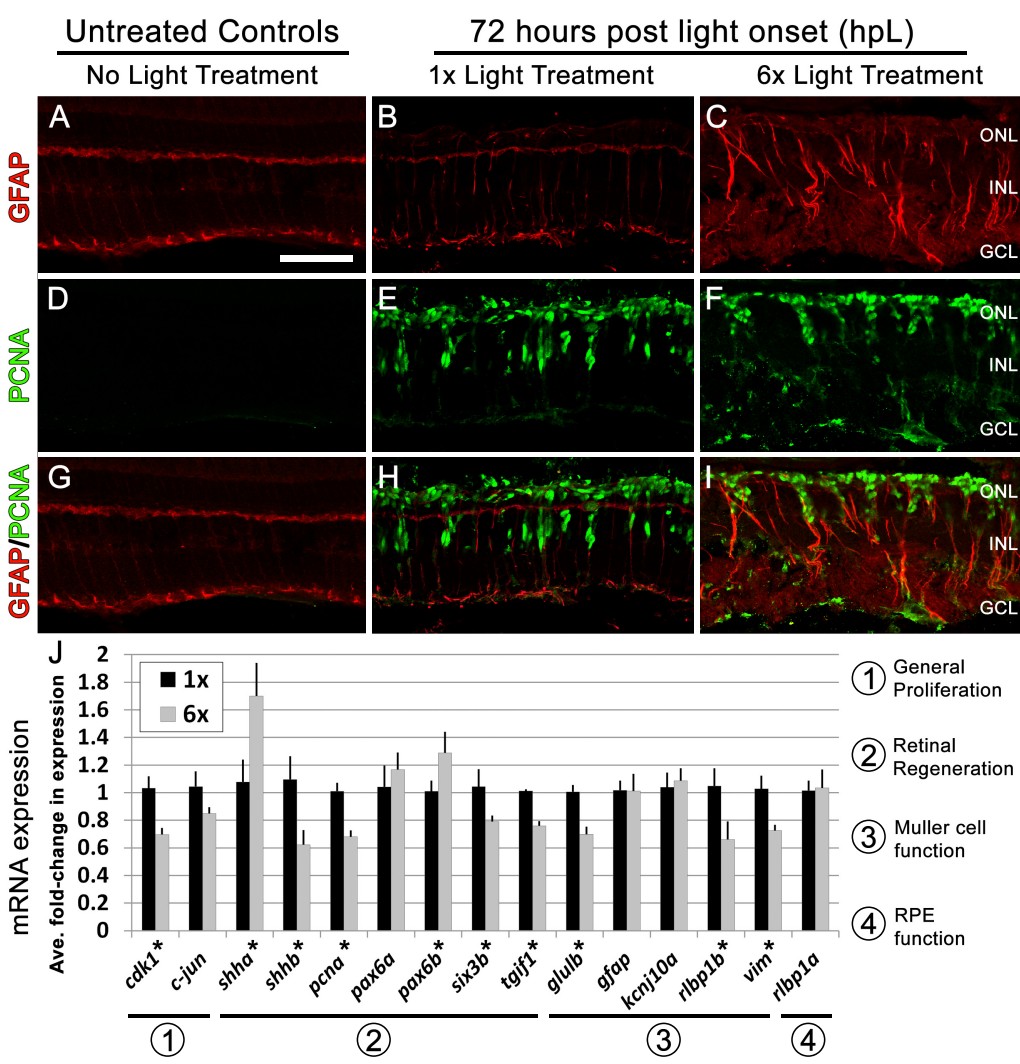

**Figure 4** **Following six rounds of light treatment Müller glia remain persistently gliotic at the canonical peak stage of progenitor proliferation and migration.** (A–I) Retinal sections collected at 72 hpL immunolabeled with anti-GFAP (red) and anti-PCNA (green) in untreated control (No Light Treatment), experimental control (1×Light Treatment), and experimental retinas (1×Light Treatment). (A–C) Müller glia are immunolabeled with anti-GFAP. (D–F) Müller glia that have re-entered the cell cycle are immunolabeled with anti-PCNA. (G–I) Merge of anti-GFAP and anti-PCNA immunolabeling. (J) Graph showing the average fold-change in expression of genes associated with (1) general proliferation (*cdk1, c-jun*), (2) retinal regeneration (*shha, shhb, pcna, pax6a, pax6b, six3b* and *tgif1*), (3) Müller cell function (*glulb, gfap, kcnj10a, rlbp1b, vim*), and (4) RPE function (*rlbp1a*). Asterisk indicates significantly different from control ($p < 0.05$). Scale bar represents 25 μm.

In addition, we observed a significantly greater number of HuC/D-positive cells in the inner plexiform layer (Fig. 5F,G arrowhead; $p < 0.05$) and outer retinal layer (Figs. 5F, 5G; arrow; $p < 0.05$) in the 6× retinas compared with the 1× retinas. This is consistent with reports demonstrating that Müller glial cells can produce excess neurons even after one round of regeneration (*Hitchcock & Raymond, 1992*; *Powell et al., 2016*; *Sherpa et al., 2008*; *Sherpa et al., 2014*).

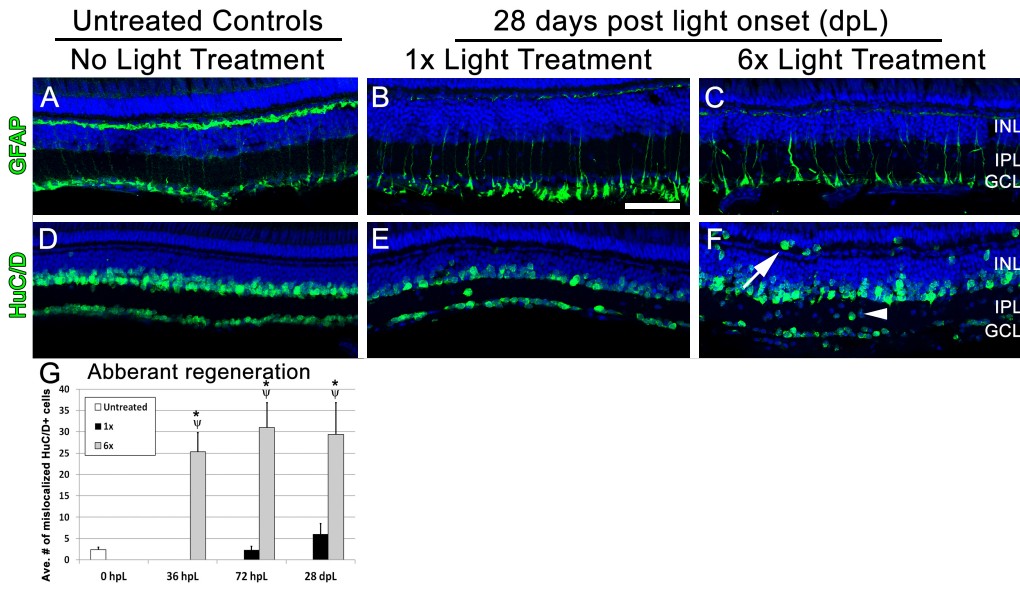

**Figure 5  Following multiple rounds of regeneration, retinas exhibit abnormal localization of inner retinal neurons.** (A–F) Retinal sections collected at 28 dpL immunolabeled with anti-GFAP and anti-HuC/D to show Müller glia and inner retinal neurons, respectively, in untreated control (No Light Treatment), experimental control (1×Light Treatment) and experimental retinas (1×Light Treatment). Nuclei are stained in blue with TO-PRO-3. (A–C) Müller glia are immunolabeled with anti-GFAP. (D–F) Amacrine and ganglion cells are immunolabeled with anti-HuC/D. Mis-localization of HuC/D-positive cells were observed in the inner plexiform layer (IPL; arrowhead) and outer retina (arrow). (G) Quantification of average numbers of HuC/D-positive cells found in the IPL and outer retina ($n = 5$ per group). Cells were counted over a linear distance of 300 μm on the central dorsal retina. Asterisk indicates significantly different from 0 h control; Psi symbol indicates significantly different from 1×retinas ($p < 0.05$). Scale bar represents 25 μm.

Lastly, we compared photoreceptor numbers in untreated control retinas with 1× and 6× retinas at 28 dpL by immunolabeling the rod and cone photoreceptor populations with cell-specific antibodies. Immunolocalization of rhodopsin (rods), Zpr-1 (red-green double cones), blue opsin (long single cones), or UV opsin (short single cones) showed that retinas subjected to 1× and 6× rounds of light treatment regenerated both rods and cones to levels observed in undamaged control retinas (Fig. 6). In contrast, blue cones exhibited reduced regeneration in the 6× retinas, but not the 1× (Fig. 6O, $p < 0.02$) and UV cones regenerated in elevated numbers in the 1×, but not the 6× retinas (Fig. 6P, $p < 0.03$). These data indicated that although six rounds of light damage results in a significant increase in the acute gliotic response to damage, the Müller glia largely retained their innate ability to regenerate photoreceptors following multiple rounds of phototoxic lesion.

## DISCUSSION

Our first hypothesis of this study predicted that upon subsequent rounds of phototoxic lesion, the exact same subset of Müller glia will not re-enter the cell cycle. We demonstrated that 35% of Müller glia re-entered the cell cycle in response to both phototoxic lesions,

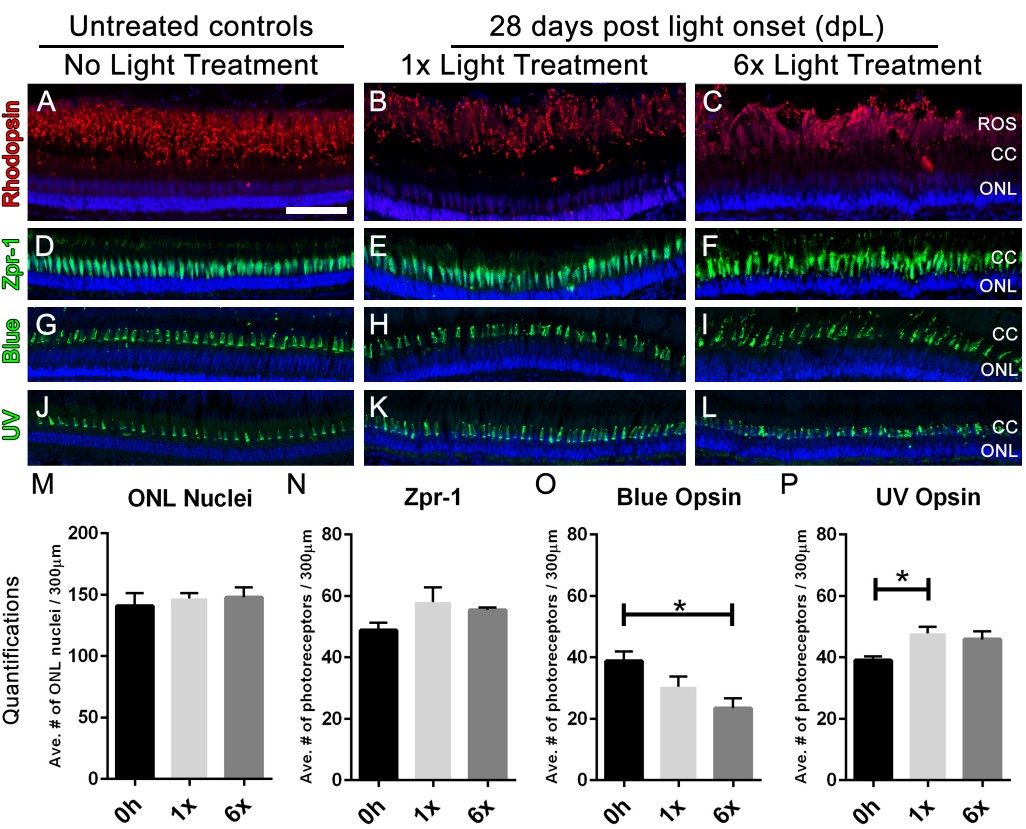

**Figure 6 Following six rounds of light treatment, retinas replace the lost photoreceptors.** (A–L) Retinal sections collected at 28 dpL immunolabeled with anti-rhodopsin, Zpr-1, anti-blue opsin, and anti-UV opsin to show regeneration of the photoreceptors in untreated control (No Light Treatment), experimental control (1×Light Treatment), and experimental retinas (1×Light Treatment) following photolytic damage. S Nuclei are stained blue with TO-PRO-3. (A–C). Rod photoreceptor outer segments are immunolabeled with anti-rhodopsin (red). (D–F) Red-green double cones are immunolabeled with Zpr-1 (green). (G–I) Long single cones are immunolabeled with anti-blue opsin (green). (J–L) Short single cones are immunolabeled with anti-UV opsin (green). (M–P) Quantification of the average number of photoreceptors 28 dpL in control and experimental groups ($n = 5$). Cells were counted over a linear distance of 300 $\mu$m on the central dorsal retina. Single asterisks indicate significant differences between groups ($p < 0.03$). Scale bar represents 25 $\mu$m.

24% of Müller glia re-entered the cell cycle only during the first round of light damage, and 40% of Müller glia re-entered the cell cycle only during the second round of phototoxic lesion (Fig. 1). This was indicated by the presence of GFAP-positive cells which were single-immunolabeled with either anti-BrdU or anti-PCNA following the second round of phototoxic lesion (Fig. 1D, arrowheads and arrows, respectively). These data suggest that the pool of Müller glia is heterogeneous, with a subset of Müller glia responding to the first round of lesion, and a different, but overlapping subset responding to the second lesion. It is possible that this serves to randomize the population of Müller glia that will respond to any given level of damage.

Our second aim of this study was to determine if there was an increase in the gliotic response of the Müller glia with additional rounds of phototoxic treatment. Zebrafish

Müller glial cells undergo a reactive gliosis response at 18 hpL, which is characterized by cellular hypertrophy in the outer nuclear layer (ONL) and physical wrapping of existing photoreceptor nuclei (*Thomas et al., 2016*). This gliotic phenomenon persists at 36 hpL, when ~50% of zebrafish Müller glial cells re-enter the cell cycle (*Thomas et al., 2016*; *Thummel et al., 2008b*), but is then down-regulated as progenitor amplification reaches its peak at 72 hpL (*Thummel et al., 2008a*). Thus, similar to what is observed during zebrafish cardiac regeneration, the onset of proliferation occurs in a gliotic/fibrotic environment and only after a large number of progenitors are present does the tissue downregulate this reactive state (*Gonzalez-Rosa et al., 2011*; *Schnabel et al., 2011*). Here we show that retinas that underwent 6× rounds of light treatment exhibited a gliotic response that was phenotypically distinct from retinas that underwent 1× round of damage. First, there was a significant upregulation of *gfap* expression in 6× retinas at 36 hpL (Fig. 3K). In addition, there was an obvious dysmorphology in the 6× Müller glia at both 36 and 72 hpL as indicated by GFAP and Glutamine Synthetase immunolabelling (Figs. 3 and 4). Specifically, GFAP immunolocalization appears to shift from being largely restricted to apical and basal endfeet of Müller glia in untreated control retinas, to increasingly more centralized expression in thick, disorganized fibers in the 1× and 6× retinas (Figs. 4A–4C). Since *gfap* mRNA expression in 6× retinas at 72 hpL is comparable to 1× retinas (Fig. 4J), this shift in immunolocalization may represent a redistribution of existing GFAP filaments needed to support hypertrophied cells.

We also observed an upregulation in *stat3* mRNA expression at 36 hpL in 6× retinas (Fig. 3K), which is indicative of an inflammatory stress response (*Gorsuch & Hyde, 2014*; *Nicolas et al., 2013*), and may be associated with the increased numbers of 4C4-positive microglia/macrophages at this time point (Fig. 2P). Microglia are known to release pro-inflammatory factors such as IL-6 that activate Stat3 in response to CNS damage (reviewed in *Smith et al., 2012*). 4C4-positive cells, which are resident and distributed throughout undamaged retinas, have previously been shown to proliferate and migrate from their existing locations to an area of retinal damage (*Craig, Calinescu & Hitchcock, 2008*; *Raymond et al., 2006*; *Wu et al., 2001*). At this point it is unclear as to whether the increased number of 4C4-positive cells observed in the 6× retinas were a result of increased proliferation of existing microglia/macrophages or whether following each additional round of phototoxic lesion the population of resident 4C4-positive microglia/macrophages increased and remained in the retina.

A previous report comparing different intensities of phototoxic lesions demonstrated that the greater the amount of photoreceptor loss, the more Müller glia re-enter the cell cycle (*Thomas et al., 2012*). Interestingly, here we observed that while photoreceptor loss was more severe at 36 hpL in retinas that underwent 6× rounds of light treatment, the proliferation response did not appear to be subsequently increased in these retinas, as indicated by similar numbers of PCNA-positive Müller glia in both the 1× and 6× retinas (Figs. 3B–3C, 3J). These data could suggest that there is a ceiling with respect to how much cell proliferation can be induced by light damage. However, we also observed a complex re-patterning of cell signaling programs in Müller glia responding to retinas that have been exposed to multiple rounds of phototoxic lesion compared to those that have only

undergone one round of damage. For example, we observed a downregulation of many pro-proliferative genes in 6× retinas at 72 hpL, including *cdk1*, *pcna*, and *shhb* (Fig. 4J), and additional signs of impaired Müller glial cell function (*vim*, *glulB*, *rlbpB*) at 72 hpL in 6× retinas compared to 1× (Fig. 4J).

In addition to these early abnormalities, we also observed aberrant localization of inner retinal neurons in the inner plexiform layer (IPL) and in the outer retina at 28 dpL in retinas that underwent 6 rounds of light damage (Figs. 5D–5F). Because BrdU labeling was not performed throughout these experiments, we cannot determine when they arose or whether they arose from aberrant differentiation of retinal progenitors or from aberrant migration of existing inner retinal neurons. However, these data are consistent with a recent report comparing multiple retinal damage paradigms that found that Müller glia regenerate excess neurons during the regenerative process (*Powell et al., 2016*). Indeed, we note that there appears to be a trend of slow build-up of these aberrantly localized retinal neurons with increasing amount of phototoxic lesion when you compare the 0, 1, and 6× retinas (Fig. 5G). Thus, while it is likely that the extent of cell mis-localization gradually builds with each damage event, additional studies are required to validate this hypothesis. This is an interesting finding because inner retinal neurons are not damaged by the light lesion, so signals to regenerate inner retinal neurons should not be present. However, we noted a significant upregulation of *shha* at the 72 hpL time point (Fig. 4J). In addition to being associated with late proliferation of the retinal progenitors, Shha is required for the development of amacrine and ganglion cells (*Neumann & Nuesslein-Volhard, 2000*). Thus, an excess of Shha signaling at this time point may erroneously drive a subset of the progenitors toward an amacrine or ganglion cell fate.

We terminated the study following 6 rounds of light treatment and 28 day recovery period so as not to exceed the normal age window of 9–18 months that we use for light lesion studies. However, we acknowledge that the age of the 6× treated fish (18 months vs. 9 months in 1× fish; Fig. S1) may be a confounding factor in our findings, including gene expression, gliosis, and aberrancy in regeneration. Indeed, to our knowledge, age has never been directly tested as a confounding factor in retinal regeneration studies. Therefore, further studies are ongoing in our lab to investigate the aging Müller glia response to retinal damage and to clearly define for the field a window of adulthood within which age is not a factor in retinal regeneration. In other tissues, such as the zebrafish telencephalon, it has been reported that the regenerative capacity of the radial glia, a comparable cell type to the Müller glia, declines with advanced age (*Edelmann et al., 2013*). This group noted that there is a decrease in the number of dividing maintenance radial glia with a concurrent increase in cell cycle time in 18 month old fish as compared to 4.5 month old fish at baseline. In response to injury, fewer radial glia re-entered the cell cycle to initiate regeneration in 28 month old fish compared to 8 month old fish (*Edelmann et al., 2013*). These experiments could indicate that the stem cell populations of the CNS appear to exhaust with age. In contrast, we observed no significant difference in the percentage of Müller glia cells that re-entered the cell cycle between the 1× (9 month) and 6× (18 month) groups (Fig. 3J). Regarding the inflammatory response to regeneration, it was previously shown that aged zebrafish fail to recruit 4C4-positive microglia/macrophages to the site of a demyelinating

injury of the optic nerve and that the quality of subsequent re-myelination is significantly reduced in aged animals (*Munzel et al., 2014*). In our experiments, the 6× treatment group shows a significantly higher number of 4C4-positive cells in the retina. However, this finding could be a result of a slow build-up of 4C4-positive cells following each round of damage and does not rule out the possibility that fewer and fewer 4C4-positive cells are recruited with each subsequent round.

Despite the increase in gliosis and aberrancies seen in recovery after phototoxic lesion, we determined that Müller glia retain the ability to re-enter the cell cycle and regenerate photoreceptors following multiple rounds of phototoxic lesion. The increased acute gliotic response of the Müller glia in retinas that were exposed to multiple rounds of phototoxic lesion may indeed suggest that they are undergoing remodeling following each round of damage and thus take on a different response phenotype as the rounds of damage increase. However, our findings highlight the incredible capacity of the Müller glia to participate in the regenerative response even in a gliotic state. In addition, to our knowledge, this is the first description of the response of the zebrafish retinal Müller glia to multiple rounds of phototoxic lesion. These findings open new avenues for future investigation into genetic programs that alter the function of the Müller cell gliotic response, and perhaps, elucidate parallels in the mammalian system that help us to better understand the delicate balance between gliosis and stemness of this important retinal cell population.

## ACKNOWLEDGEMENTS

The authors would like to thank Xixia Luo for excellent fish husbandry and technical support.

### Funding

This work was funded by the National Institutes of Health grants P30EY04068 (Anatomy) and R01EY026551 (Ryan Thummel), and start-up funds to Ryan Thummel, including an unrestricted grant from Research to Prevent Blindness to the Wayne State University, Department of Ophthalmology. Jennifer Perez was supported by a Thomas C. Rumble Fellowship provided by the Wayne State University Graduate School. The funders had no role in study design, data collection and analysis, decision to publish, or preparation of the manuscript.

### Grant Disclosures

The following grant information was disclosed by the authors:
National Institutes of Health: P30EY04068, R01EY026551.
Start-up funds.
Wayne State University, Department of Ophthalmology.
Wayne State University Graduate School.

## Competing Interests

The authors declare there are no competing interests.

## Author Contributions

- Alexandra H. Ranski and Ashley C. Kramer conceived and designed the experiments, performed the experiments, analyzed the data, authored or reviewed drafts of the paper, approved the final draft.
- Gregory W. Morgan performed the experiments, analyzed the data, approved the final draft.
- Jennifer L. Perez performed the experiments, approved the final draft.
- Ryan Thummel conceived and designed the experiments, analyzed the data, contributed reagents/materials/analysis tools, prepared figures and/or tables, authored or reviewed drafts of the paper, approved the final draft.

## Animal Ethics

The following information was supplied relating to ethical approvals (i.e., approving body and any reference numbers):

The Institutional Animal Care and Use Committee at Wayne State University approved the phototoxic lesion protocol used in this study (Protocol # 16-01-037).

## Data Availability

The raw data are provided in the Supplemental Files.

## Supplemental Information

Supplemental information for this article can be found online at http://dx.doi.org/10.7717/peerj.5646#supplemental-information.

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
