# Peer review of "Characterization of retinal regeneration in adult zebrafish following multiple rounds of phototoxic lesion"

_PeerJ, doi:10.7717/peerj.5646_

## Round 0.1 · original submission · Major Revisions

Thank you for submitting your work to PeerJ. You will find three thorough reviews of your manuscript, with two reviewers recommending Minor Revisions and a third recommending Rejection. I have decided on the recommendation of Major Revision and invite you to respond to the suggestions of each reviewer and resubmit your manuscript. While all reviewers recognized value in this study there were some significant concerns raised about experimental design and support for the conclusions.

While I ask that you respond to all reviewer comments, there are several areas that appear to be most significant:

1. Please address reviewer comments that you may be over-generalizing the mechanisms behind tissue regeneration in different tissue types.

2. The introduction makes a claim about different subsets of Muller glia reentering the cell cycle that is not presented in the results section.

3. A clear hypothesis could be presented in the introduction, with a rationale for why six rounds of light exposure was chosen.

4. Make clearer connections between results statements and conclusions and the data presented. Please address the concerns of reviewer 3 in particular that the conclusions are not adequately supported by the presented data and that since regeneration is not examined after each round of light exposure it is not known whether six rounds of regeneration took place.

5. Additional information is needed in the methods section for various experiments, such as the immunofluorescence and qRT-CPR. For the latter I would suggest reviewing the MIQE guidelines for minimal information needed (Bustin et al. 2009, Clinical Chemistry). Please describe if tissues were frozen or stored prior to processing, whether DNase treatment was done, the primers used and if they spanned introns, whether qPCR products were sequenced to confirm their identity, what endogenous control genes were used, how fold changes in expression were calculated and what statistic was used.

6. Additional quantification was suggested for various results. Please also address comments about lack of some controls.

7. There are typos that need correcting.

Reviewer 1 ·

Basic reporting

This study by Ranski et al. compares photoreceptor regeneration and Muller glial activation in zebrafish retinas subjected to one versus six rounds of acute light damage and regeneration. Although studies have been done to determine the regenerative ability of the zebrafish fin after repeated rounds of injury, this has not been examined for zebrafish retina. Therefore, these data are of interest to the field. The study is straightforward, the methods and results are mostly clearly explained, and the conclusions seem reasonable. To me, the most interesting result is that repeated rounds of light damage appear to have a destabilizing effect in the retina, although the authors do not highlight this. I have only a few minor suggestions to improve the manuscript.


Introduction, line 57: in comparing mechanisms of regeneration of various tissues such as the heart, retina, and the fin in the zebrafish, the authors state that there are “resident cells in each tissue that have the capacity to act as stem cells…” In fact, it is my understanding that in the fin, several studies have shown that regeneration does not result from activation of a resident stem cell, but rather from the dedifferentiation of mature (non-dividng) cells, each of which are lineage restricted (i.e., osteoblasts only regenerate bone, dermal fibroblasts only regenerate epidermis, and so on). This is clearly not the same mechanism as is described for retinal regeneration in zebrafish. The authors should therefore take care not to over-generalize about mechanisms of regeneration across tissue types.

Introduction, lines 75-80: the authors again attempt to ascribe parallel mechanisms of regeneration in the retina, heart, and fin by suggesting that following injury each tissue participates in the initial formation of a scar, followed by scar regression and then cell proliferation/differentiation. However, this is also an over-generalization. In heart and skin regeneration-associated scarring occurs via fibroblast recruitment and deposition of cartilage. However, in retinal damage any scarring would be formed by the Muller glia themselves. Although zebrafish Muller glia undergo reactive gliosis in response to retinal damage, they do not produce a scar prior to initiating regeneration. This difference with heart and fin could be related to different regenerative mechanisms needed for neural as opposed to non-neural tissue. In any case, care should be taken not to draw unsupported parallels here.

Introduction, lines 96-99: “We found that a different subset of Müller glia reentered the cell cycle during the second round of regeneration…” These experiments are not described in the Results, although that is a very interesting finding!

There are several typos throughout the manuscript; the corrections are listed below:

The title should read “Characterization of retinal regeneration in adult zebrafish following multiple rounds of degeneration and regeneration”

Abstract, line 36: “suggests that all vertebrate Müller glia have the potential to act as…”

Abstract, line 40: “We observed that Müller glia to re-entered the cell cycle…”

Abstract, line 47: “Finally, we following the sixth round of regeneration, we…”

Introduction, line 74: “photoreceptor ablation by intense light…”

Introduction, line 94 (and elsewhere): I suggest using the phrase “dark-adapted” instead of “dark-treated”

Discussion, line 316: “Overall, our findings results demonstrate that zebrafish can regenerate…”

All figures: “immunolabeled by…” should be anti-rhodopsin, anti-UV opsin, anti-blue opsin, anti-PCNA, etc. (except for monoclonal antibodies with their own names, such as Zpr-1, 4C4, etc.).

All Figure legends: what length does the scale bar represent?

Experimental design

Materials and Methods, Photoreceptor Lesions: Please include information on how many fish were in each group, and how many biological replicates were performed for each light damage experiment (1x vs. 6x)

Results, lines 190-193 and Figure 1: The authors state that there was no difference in the number of regenerated rod photoreceptors after either 1 or 6 rounds of light damage, but no quantification is provided. Please include these data. Also, in the schematic at the top of Figure 1, shouldn’t the text in between the two timelines read “5 additional rounds”?

Validity of the findings

Figure 1: The DAPI staining at 6x compared to 1x rounds reveals a fair amount of disorganization in the ONL with a suggestion of incipient rosettes, indicating that repeated rounds of light damage is destabilizing the retina (consistent with the persistent activation of the Muller glia shown in Fig. 3). It would be nice if the authors commented on this in the results and discussion.

Results, lines 198-202 and Figure 2: Why is there so much residual rhodopsin staining at 36 hpL in the 1x image? Shouldn't these cells be almost completely ablated in the dorsal retina? The presence of additional 4C4+ microglia in the 6x retina is also interesting: are these "extra" microglia that continue to hang around between the multiple rounds of light damage, or do they go away and re-appear in greater numbers each time? Please comment on this (or if you have IHC data on 4C4 at 28 dpL please show that). Also, there is again no quantification shown (or provided in the dataset) for the rods.

Results, lines 214-215 and Figure 4: Looking at the extent of the GFAP-labeled Muller glia processes in panels 4B-4B”, is appears that the 6x regenerated retinas are thicker than 1x. Is this the case? If so, does this suggest an accumulated effect of over-proliferation with each round of regeneration?

Discussion, lines 257-266: Another possible explanation for the results is that each round of degeneration/regeneration weakens or destabilizes the retina a little more, lowering the resistance to subsequent injury (and thus undergoing more damage). This might be indicated by the persistently activated Muller glia, the disorganized appearance of the ONL, and the increased number of 4C4+ cells in the 6x retina.

Discussion, line 282: As stated above for the Introduction, I don’t think it is accurate to term this a “scar-like response”

Discussion: the authors should comment on the apparent discrepancy between the reduction in gfap mRNA expression and the increased GFAP protein in 6x retinas. What is known about the post-transcriptional regulation of GFAP expression?

Additional comments

Please see attached review

·

Basic reporting

This is a great study to add to our understanding of the extent to which zebrafish can regenerate neurons. It is a clearly written manuscript with a coherent story provided about the ability of adult zebrafish to regenerate retinal neurons after multiple rounds of injury.

Some of the introduction material could be reorganized to make the comparison between regeneration in different tissues and the retina more apparent. For example, a previous paper (Qin et al, 2009, PNAS) showed that mps1 is needed in retinal regeneration, and that there may shared mechanisms of epigenetic regulation with the fin. This would be a good reference to include in the introduction section.

The authors should consider stating a clear hypothesis at the close of the introduction section. Based on the introduction, I would presume that the study was aimed at determining if there is a scarring reaction that is then mitigated during regeneration similar to other tissues, but that is not clearly state. Additionally, the assertion at the end of the introduction that different subsets of Müller glial cells reenter the cell cycle was not apparent in the results presented in the manuscript.

The figures are all directly relevant to the goal of the study. Figure 4C might be easier to interpret for the reader if the genes were grouped to reflect the results text and discussion. For example, a graph with proliferation-related genes and a graph of intermediate filaments, etc., or alternatively group them within one graph. This would make interpreting the data more straightforward. The figure legends should be reworded to be clear that cells aren’t labeled with a protein, but rather with an antibody. (I.e. “Rod photoreceptor outer segments are immunolabeled with anti-rhodopsin” instead of “…with rhodopsin”.)

The statistics reported on line 193 do not have any matching quantitative data, so this should be resolved.

There are some language conventions that make the manuscript feel a bit disjointed as a whole. For example, in the long second paragraph of the introduction there is a string of sentences that all begin with single transition words that feels very redundant. In lines 285-287, the pronoun reference for ‘this’ is confusing.

The authors refer to “Müller cell disorganization”. It also appears that lamination of the retina is disturbed at 36 and 72 hours post light onset in the 6x treatment, possibly also with a reduction in the number nuclei in the outer nuclear layer. Is this consistent across biological replicates observed or just appears so in the images in the figures?

Experimental design

More detail on the immunofluorescence analysis should be included. In particular: 1. Include the number of biological replicates that were examined per treatment and timepoint, and 2. State whether the data presenting average number of cells was over a particular linear distance (figs 2K and 5E).

Because the immunofluorescence was not done in a quantitative manner, I suggest the authors be cautious in making assertions about the relative levels of expression of proteins. If, however, images were taken in such a way to ensure valid comparisons of the level of fluorescence across groups, then these methods should be clearly described. For example in Figure 3 A-B, one could argue that the GFAP might just be redistributed in the 6x treatment, as the end feet of the Müller glia in the 6x treatment, particularly at the apical side, appear to have lower GFAP amounts relative to the 1x treatment.

Was retinal tissue used for qRT-PCR put directly into Trizol when dissected or frozen for storage (or other)?

Validity of the findings

In general, I think the Discussion section could be much stronger and better serve the findings of this study. It is not clear how the proposed explanations in the first paragraph of the discussion were developed. If there are references to support these hypotheses, they should be included. The discussion should better reflect the Introduction that set up the idea that regenerative responses start with a scarring response, and later the scar is replaced by regenerated tissue. The authors provide clear evidence that there is a much stronger gliotic response from Müller glia in the 6x treatment and that at 28 days post light onset the retinal structure appears normal. This would fit quite well with the information in the Introduction. The final closing paragraph should better state the benefit this study provides to the field.

Additional comments

This is a great study that many will find interesting and valuable. With minor revisions to the text and clarification on some of the methods, the presentation of the data will be even more impactful.

Reviewer 3 ·

Basic reporting

This study provides some interesting observations related to photoreceptor regeneration after damage due to light treatment in zebrafish, through the use of repeated light treatments and recovery periods.

Although the concept of repeated damage – and what are the consequences? – is intriguing, the manuscript offers very little in the way of new and important information regarding retinal regeneration. More importantly, this manuscript is missing several key controls, makes numerous unjustified assumptions regarding the outcomes of the experiments, and as a consequence greatly overstates the results. Very few of the authors’ claims are adequately supported by evidence.

The manuscript does not read well and is in need of proofreading. For example, line 47 contains many typos: “Finally, we following the six round of regeneration.” In lines 118-119, do the authors mean “photolytic” rather than “photoletic?” Photoletic does not appear to be a word. Line 143, was this really an antibody directed against UV light? In line 272, “phenomenon” is the singular form of “phenomena.”

Methods, line 112. What ages/sizes of fish were used?

Methods, line 127. Clarify whether the age-matched adults underwent the one round of damage (1X) at the same time/age as when the 6X group underwent the sixth round. The diagram at the top of Figure 1 does not convey this information very well, but implies that the 1X were sampled dozens of days before the 6X were sampled.

Methods, line 168. More detail is needed for the quantification methods. Were cells counted from specific, defined regions of a section? From all retinal layers? What was the denominator for the counts – per section, per a particular distance of retinal section? Sample sizes for each study should be provided here as well.

Methods, lines 169-171. The use of ANOVA and then T-test makes no sense at all. For multiple groups, ANOVAs and an appropriate post-hoc test should be used.

Methods, line 173. This section must describe the qPCR primers, in a Table or similar. Where are they?

Methods, line 181. Should provide a brief description of quantification methods for qPCR analysis. What were the internal standards/control transcripts and primers?

Results, line 255. Ectopic and supernumerary neurons in regenerated retina have been shown by many others, including but not limited to Hitchcock et al., 1992, Sherpa et al., 2008, and Sherpa et al., 2014.

Discussion, line 313. This reference does not show Shh function in development of amacrine and ganglion cells. Did the authors mean Neumann et al., 2000, or perhaps Shkumatava et al., 2004?

Figure 2 legend, line 344. Do the arrows point to individual cone opsins? Why is opsin capitalized?

Experimental design

This study provides some interesting observations related to photoreceptor regeneration after damage due to light treatment in zebrafish, through the use of repeated light treatments and recovery periods.

Although the concept of repeated damage – and what are the consequences? – is intriguing, the manuscript offers very little in the way of new and important information regarding retinal regeneration. More importantly, this manuscript is missing several key controls, makes numerous unjustified assumptions regarding the outcomes of the experiments, and as a consequence greatly overstates the results. Very few of the authors’ claims are adequately supported by evidence.

Major concerns:

It is not clear why the authors carried out these studies. The introduction lays out an interesting context of scar formation vs. regeneration, but the experiments themselves are not clearly related to this context. There is no obvious rationale for the experimental design. Why were six rounds selected as the experimental design? Why not 3? Why not 10?

The authors did not demonstrate that multiple rounds of degeneration and regeneration were actually accomplished, and with a time-course similar to the single round. This is merely assumed based upon information derived from prior work using the procedure for one round of degeneration and regeneration. It is possible that each round of light treatment results in different levels of damage. It is also possible that the response in each round has different kinetics. The “peak level” of proliferation sampling time (72 hpL) in the 6X group may not be the same as in the 1X group, and so the comparison of gene expression, etc. at this time is completely unjustified. The study, therefore, can only claim multiple rounds of high intensity light exposure, not regeneration. Collectively these concerns constitute a significant shortcoming in this manuscript.

Quantification of data is not clearly described, making it difficult to know what was actually counted and compared. Furthermore, no undamaged controls are included, and so true comparisons across samples are not even possible.

Specific comments:

Title: The title is not consistent with what is actually shown in the manuscript.

Abstract: The abstract also contains information not supported by the findings – six rounds of degeneration/regeneration were not demonstrated.

Abstract conclusions should be changed to say that increased numbers of mislocalized HuC/D cells were found. As is, it suggests that 1X light damage does not result in this feature, but it does based on their presented data. And line 103-104 of intro should be changed accordingly.

Author contributions: “AR performed, collected…. Except where noted.” Noted where?

Introduction, line 97. Where did the authors study different subsets of Muller glia?

Results, line 197. What is “key” about the time points of 36 and 72 hpL?

Results, lines 203-204. What is the rationale for the microglial investigation, and the meaning of the microglial data? This finding is not linked to any particular concept here. In the parentheses, what do the numbers refer to (13.8 vs. 46.2)? Are these #s of microglia per retina? Per section? What?

Results, line 206 and nearby. Why was 36 hpL selected, and how is it known that this is a valid comparison for the 6X samples?

Results, line 214. The peak of progenitor amplification may be known from prior 1X experiments, but this is not demonstrated for the 6X experiments.

Discussion, line 258. What is meant by, “sensitive?”

Discussion, line 261. Is it true that phototoxicity is opsin-dependent? Has this actually been demonstrated for any model organism? Also, it would be easy to test this explanation by measuring opsin expression.

Discussion, line 288. Are microglia known to release pro-inflammatory cytokines in this particular context?

Discussion, line 296-297. Please include citations for each of the genes stated to be involved with each process/function.

Discussion, line 291. How do the authors know this is the peak time of proliferation and migration in the 6X example? This is an assumption based upon what is known of the 1X situation. 72 hpL may not be the peak at all.

Discussion beginning in line 307. The authors did not demonstrate a slow build-up of mislocalized neurons in Figure 5. If they have indeed done these studies (data not shown stated), then these data should be included in regards to comments above. Statements should be edited accordingly.

Figure 1 legend. The first figure should really be related to damage rather than “regeneration,” to demonstrate that the approach used actually accomplishes the intended outcome. And again, there is a need to demonstrate that each round causes damage – and is it the same sort of damage? Also need to demonstrate actual regeneration in each round in order to use this terminology.

Figure 1. The diagram is misleading if the animals (1X vs. 6X) were actually age-matched.

Figure 1, continued. Why are there no undamaged controls for comparison?

Figure 1, continued. A conclusion based on quantification is stated in the text along with p > 0.05. However, quantification analysis is not presented. Was a quantification of photoreceptor types performed and if so, how? If not, it should be done. This claim forms a basis for the presented study and is an important conclusion (discussion line 317, intro line 100). Further, this conclusion is stated in the text but is missing from the abstract.

Figure 2. The data in this figure needs to include TUNEL stain with quantification. Ideally, co-label and quantification of TUNEL and each of the photoreceptor cell types. As mentioned above, this should probably constitute the first figure. With the increased number of 4C4+ cells found at the site of light damage, it would be interesting to see if they are all associated with TUNEL in some way? Since the antigen for 4C4 is unknown, it is not clear if this antibody marks only microglia or if it also marks macrophages. Text should be changed accordingly. Also, without demonstrating photoreceptor death at each exposure to intense light, it is not possible to interpret the 4C4 images.

Figure 2. In the graph (K), zpr1+ quantification appears equal between 1X and 6X light damage, however the images shown suggest increased numbers of zpr1 present in 6X light damage. If the quantification is correct, then better representative images should be chosen.

Figure 2, continued. In K, the Y axis is not fully labeled. Photoreceptors per what? And again, undamaged controls are needed.

Figure 3. Where are the undamaged controls for comparison?

Figure 3, continued. C’ to D’ comparisons of Muller glia features. Please provide more information about different visual features between the Muller glia in 1X to 6X. This figure appears to be the basis for the hypertrophy conclusions stated in the abstract, however, this hypertrophy appears transient since the images at 72 hpL and in regenerated retinas (Figure 5) do not reveal Muller glia hypertrophy. The conclusions and abstract should be edited accordingly to specify that this is a transient feature.

Figure 3F. qPCR and PCNA stains suggest increase stat3 without subsequent increase in cell cycle. Please discuss this further and it’s implications, since one function of stat proteins is to induce cell proliferation. Also, what are the transcriptional outcomes of stat3 signaling in Muller glia?

Figure 4. Need undamaged controls for comparison.

Figure 4, continued. Muller glia endfeet appear different in B’’ to A’’, also noted in Figure 3 comparing C to D. 4C: for presentation purposes, helpful to group and annotate the genes by process to facilitate reader comprehension.

Figures 3 and 4. Upregulation of gfap also seems to be transient at 36 hpL (figure 3) as it is not different at 72 hpL (Figure 4). Conclusions and relevant statements should be edited accordingly in abstract and text.

All Figures containing qPCR data. The format for data presentation is rather strange, and cluttered by the inclusion of all of the 1X (which are all, ”1” when using the Fold Change as the Y-axis). Fold-change for each (1X and 6X), compared to levels in undamaged controls, may be a more useful metric, as it will show the response to the treatment in the 1X case vs. the 6X case. However, this reviewer remains skeptical that these time points are truly comparable, because the kinetics of the response to light treatment may be different in the 1X vs. 6X group.

Figure 5. Need undamaged controls for comparison. In E, the Y axis is incompletely labeled. # HuC/D+ cells per what?

Figure 5, continued. Why focus only on HuC/D+ neurons? Were photoreceptors mislocalized also, as seen after a ouabain lesion (Sherpa et al., 2014)? There are non-Hu+ nuclei in the IPL.

Validity of the findings

This study provides some interesting observations related to photoreceptor regeneration after damage due to light treatment in zebrafish, through the use of repeated light treatments and recovery periods.

Although the concept of repeated damage – and what are the consequences? – is intriguing, the manuscript offers very little in the way of new and important information regarding retinal regeneration. More importantly, this manuscript is missing several key controls, makes numerous unjustified assumptions regarding the outcomes of the experiments, and as a consequence greatly overstates the results. Very few of the authors’ claims are adequately supported by evidence.

Major concerns:

The authors did not demonstrate that multiple rounds of degeneration and regeneration were actually accomplished, and with a time-course similar to the single round. This is merely assumed based upon information derived from prior work using the procedure for one round of degeneration and regeneration. It is possible that each round of light treatment results in different levels of damage. It is also possible that the response in each round has different kinetics. The “peak level” of proliferation sampling time (72 hpL) in the 6X group may not be the same as in the 1X group, and so the comparison of gene expression, etc. at this time is completely unjustified. The study, therefore, can only claim multiple rounds of high intensity light exposure, not regeneration. Collectively these concerns constitute a significant shortcoming in this manuscript.

Results, line 188. How do the authors know there was regeneration in each round? The terms, "light treatment and recovery" might be better, with an assumption of regeneration based upon the outcome after 6 rounds of damage. Even then there are still major unknowns regarding kinetics of any damage and/or any regeneration, such that there is little or no justification for the selection of sampling times.

---

## Round 0.2 · Major Revisions

Thank you for submitting a revision to your manuscript. All three reviewers agree that this new version is greatly improved. Two of the reviewers have some minor comments to address that request some edits. The third reviewer has more extensive comments that should be addressed. Most of these are editorial comments, although they have requested supplemental data comparing 9 month and 18 month 1X light treatment samples. These data are referenced in the rebuttal letter, with an example shown. Please consider the reviewer’s request that these data be added as a supplement to support the direct comparison of fishes of different ages, and to support the discussion, lines 410-434.

Reviewer 3 also expresses concern with the qPCR data presented in figure panels 3K and 4J, as the age of the fish is not controlled for. They recommend a new qPCR design that includes age-matched undamaged embryos for the 1X and 6X treatments. Recognizing that this is would be a substantial amount of work, I would recommend either removing these qPCR data, or clearly stating in the discussion a caution about the lack of age-matched controls. Are there previous comparisons of gene expression in 9 and 18 month old embryos that could support the conclusion that these genes do not change expression with age?

Thank you for the added details on the qPCR methods. Line 209 states that “about” 1 microgram of RNA was used in the reactions. Is there a reason for this approximation, or was 1 microgram used in each cDNA synthesis reaction? Since volumes of the cDNA reaction are not given, can you add the RNA equivalent that was used in each PCR reaction (the equivalent mass of RNA used in the 2 microliters of sample)? You have added that PCR products were not sequenced to confirm identity. Where they run on a gel to confirm that products were of predicted size? Was the melt curve analysis used to confirm that single products were produced? These details would be helpful, along with the type of statistical analysis used and efficiencies for each primer set. Lastly, while shown as being statistically significant, many of the fold changes in expression in panel 4J seem small. Are there any studies on these genes that would indicate what is considered functionally relevant changes in expression?

Please check that your uploaded figures include the revised figure legends. These revised legends appear in the main text, but are not part of the generated figures in your review PDF.

I look forward to seeing your revised manuscript.

Reviewer 1 ·

Basic reporting

This revised manuscript by Ranski et al. is much improved over the original submission and most of my previous concerns have been addressed. In particular, the addition of data for Muller glia proliferation after one vs. two rounds of light damage, better descriptions of the methods, and revision of text in the Introduction and Discussion significantly clarify and strengthen the study. I just a have a few minor changes to recommend prior to publication:

1) Methods, line 156. I’m not sure what the sentence “The null hypothesis was accepted…” is referring to.
2) Results, lines 243-244 and Figure 2. The interpretation that “the same population of Muller glia do not re-enter the cell cycle following each individual light damage event” is not accurate. If 35% of the MG are double labeled, this is in fact a significant proportion of cells that do re-enter the cell cycle. And looking at the quantification of the data in Figure 2, it seems that there is not significantly more PCNA only cells (2nd round) than double labeled cells (1st plus 2nd round). So this indicates that in fact it is mostly the same population from the first round, with some added PCNA single positives in the second round. Therefore, the authors should revise their interpretations of these data accordingly (and should also revise how they refer to these data in the Introduction and Discussion).
3) Discussion lines 387-388: “This suggests a complex re-patterning of cell signaling programs in Müller glia responding to retinas that have been exposed to multiple rounds of phototoxic lesion…” The data also suggest that there may be a ceiling with respect to how much cell proliferation can be induced by light damage.

4) Discussion lines 435-436: “Despite aberrancies seen in recovery after phototoxic lesion, we determined that the long term regenerative potential of Müller glia appears to remain relatively intact.” I agree, but in the Introduction (lines 112-113) the authors state that it's possible the MG are beginning to exhaust that stem cell potential after multiple rounds of LD. The authors may want to pull back on that statement if they don't think there is strong evidence for it.

Experimental design

see above

Validity of the findings

see above

Additional comments

see above

·

Basic reporting

This revised manuscript is a much more coherent reporting and contextualizing of the study.

Experimental design

The methods have been greatly revised to make the approach, sample sizes, and data analysis more transparent and repeatable.

Validity of the findings

Given the revisions of the methods, figures, and results, the outcome of the study is now clearer to interpret and better put in context throughout the manuscript.

Additional comments

This resubmitted manuscript is a much more cohesive description of the study examining how multiple rounds of phototoxic damage impacts the regenerative response. The extensive revisions by the authors have greatly improved the readability and ease of data interpretation. I have one small suggestion and something for the authors to consider in ongoing research.

In line 316, the statement "...consistent with a recent report..." should probably read "...consistent with reports..." since there are multiple citations and one is quite old.

The results that there is more photoreceptor death but a comparable proliferative response in the 6x compared to the 1x treatment despite using the same damage paradigm continues to be intriguing. The revised manuscript discusses this much more clearly. It would be interesting to consider in ongoing research if reduction of Muller glial cell function, as the data in the paper indicate, reduces neuroprotection of damaged photoreceptors. That would result in less photoreceptor survival (as indicated by more photoreceptor cell death) in response to the same light damage.

Reviewer 3 ·

Basic reporting

This version of the manuscript is greatly improved, with the addition of necessary control samples to the Figures, detailed explanation of the now appropriate statistics, better terminology and referencing of the literature, and the acknowledgment of study limitations with age as a confounding variable.

However, there remain several places in the manuscript that require improvements in clarity, the authors continue to overstate findings, and there are a number of other issues.

Introduction, lines 61-65. It is not clear how the findings from Qin et al. (2009) support the conclusions stated in the last two phrases of the sentence.

Introduction, line 103. “Normal numbers” – same issue as in the abstract.

Introduction, lines 112-113. Since no epigenetic studies (as the field currently considers such studies) were performed here, this statement is not warranted.

Methods, line 156. “The null hypothesis…..” This information belongs in Results, not Methods.

Methods, line 217 (and in the description of Table 1). “span exons” Do the authors mean, span introns? Or span exon-exon junctions (splice junctions)?

Consider re-ordering Figures 1 and 2 (switching them). This would make more sense given the ptic/data in Figure 3 and those that follow.

Results, line 226. This first section would benefit from a brief summary of the characteristics of damage due to the light damage method used (e.g., types of photoreceptors destroyed, timeline, any dorsal/ventral regional differences, etc.).

Figure 1. Are these data following 2X light exposure? Needs to be clearly stated in figure legend and indicated in figure.

Results regarding Figure 1 (line 224). Header should be reworded as some of the BrdU+ MG were also PCNA+ (35% according to the authors’ data).

Results, line 266. Do the authors mean, not significantly different?

Results, line 308. “near-normal numbers” – this is even more vague and confusing than “normal numbers.”

Results, line 316. More than one report is cited, and at least one of them is definitely not recent.

Discussion, lines 340-342. This reviewer struggled to understand this sentence.

Discussion, line 369. Upregulation in stat3 at 36 hpL – in what vs. what?

Discussion, line 371. The 4C4+ microglia could also contribute to increased inflammatory response. At best in this study, they are “associated” with multiple rounds of light exposure.

Discussion, line 373. The wording of this sentence suggests that 4C4+ microglia specifically have been shown to secrete IL-6. This statement should be edited to clarify that “microglia” have been shown the release inflammatory factors, since 4C4 was not used in the cited study. Cited paper is a review, and should be noted as such.
Discussion, line 381. What are “normal circumstances?”

Discussion, lines 347-348. Same struggle here.

Discussion, lines 431-433. The first part of this sentence contradicts the second part. The sentence should be removed.

Discussion, line 442 and 447. The term, “epigenetic” in the recent literature has come to convey primarily chromatin alteration. Although the authors may intend to use the term more broadly, use of the term comes across as misleading, because chromatin was not investigated.

Experimental design

Abstract in general. The statement of gliotic scars followed by the potential of fish MG to become gliotic in abstract is still distracting, as it isn’t directly tested in this study. The abstract should instead start with a more compelling rationale for multiple rounds of light damage, as current wording of the abstract still makes this rationale unclear. Further, the claims of a gliotic response by fish MG here are not fully substantiated, as it is entirely possible that the multiple rounds of light exposure actually damage the MG directly. This should be considered throughout the manuscript and warrants editing in all sections.

Introduction, lines 88-89. Still need more justification for this study. Was the goal to see if multiple rounds of light exposure would lead to a gliotic response and/or failure to regenerate?

Methods, Results, and Figures in general. Regarding damage at each round of light exposure—it needs to be clear in the MS that this damage is assumed (based on loss/alterations of staining for photoreceptor specific markers/proteins) because no cell death was actually directly analyzed. For example, the repeated light exposure could potentially lead to opsin denaturing (ie unfolding or alteration of the antigens detected by the antibodies used). In addition, along those lines, what can be seen from the DAPI+ nuclei in INL look abnormal (Figure 2, however images are cropped). This should also lead to a consideration of their findings regarding different “subsets” of Muller glia…does repeat light damage damage the MG (would be indicated be TUNEL in INL/colocalization with GFAP/GS) which would lead to characteristics seen (such as different subsets undergoing cell division or alterations in their morphology), or are the MG truly “gliotic”? Given the above discussion, the terms ‘damage & recovery” are not warranted, instead terminology should be limited to “multiple rounds of light exposure”.

Methods, line 125. Section header “Photoreceptor Lesions” should instead be “Intense Light Exposure.”

Results, first paragraph. It seems odd that the proliferating MG were not simply quantified – in this way the authors could potentially compare 1X to 2X to 6X using PCNA.

The provided clarification of ages in regards to experimental design and sampling is essential to interpreting the study and considering any conclusions. The comparison of 9 mo to 18 mo fish for 1X light treatment data should be included as (supplemental) data. This was described in the authors’ rebuttal but not included in the revised MS.

Results, lines 282-304 and Figure 4J. The 1x vs. 6x ddCT comparison is not justified, the additional age variable is a problem here. This reviewer suggests comparisons of age-matched undamaged vs. 1x, and also age-matched undamaged vs. 6x.

Results, line 309 (and Figure 6). An explanation/justification for the selection of 28 dpL is needed. Again, it is entirely possible that the kinetics to get to this number do not match for 1x vs. 6x.

Discussion, line 359. The authors did not demonstrate six rounds of damage. Suggest the term “phototoxic treatment,” or “light exposure.”

Validity of the findings

Introduction, line 91. Regarding stem cell response of Muller glia—this was not directly tested in this MS beyond their ability to proliferate, and even that was limited.

Introduction, line 94. Why? Because the Muller glia have limited stemness or because they themselves are damaged?

Introduction, line 100. Again, did not demonstrate damage. Rather, exposing the retina multiple times to intense light.

Methods, Results, and Figures in general. Regarding damage at each round of light exposure—it needs to be clear in the MS that this damage is assumed (based on loss/alterations of staining for photoreceptor specific markers/proteins) because no cell death was actually directly analyzed. For example, the repeated light exposure could potentially lead to opsin denaturing (ie unfolding or alteration of the antigens detected by the antibodies used). In addition, along those lines, what can be seen from the DAPI+ nuclei in INL look abnormal (Figure 2, however images are cropped). This should also lead to a consideration of their findings regarding different “subsets” of Muller glia…does repeat light damage damage the MG (would be indicated be TUNEL in INL/colocalization with GFAP/GS) which would lead to characteristics seen (such as different subsets undergoing cell division or alterations in their morphology), or are the MG truly “gliotic”? Given the above discussion, the terms ‘damage & recovery” are not warranted, instead terminology should be limited to “multiple rounds of light exposure."

The provided clarification of ages in regards to experimental design and sampling is essential to interpreting the study and considering any conclusions. The comparison of 9 mo to 18 mo fish for 1X light treatment data should be included as (supplemental) data. This was described in the authors’ rebuttal but not included in the revised MS.

Results, lines 282-304 and Figure 4J. The 1x vs. 6x ddCT comparison is not justified, the additional age variable is a problem here. This reviewer suggests comparisons of age-matched undamaged vs. 1x, and also age-matched undamaged vs. 6x.

Results, line 309 (and Figure 6). An explanation/justification for the selection of 28 dpL is needed. Again, it is entirely possible that the kinetics to get to this number do not match for 1x vs. 6x.

Discussion, line 430. This study did not address recruitment of 4C4+ cells. Findings were limited to a presence of these cells and their origins remain unknown.

Discussion, last paragraph (starting line 435). This study did not demonstrate long-term regenerative potential of Muller glia. Rather, it showed that Muller glia likely re-entered cell cycle (and perhaps were reactive) following multiple rounds of light exposure. See discussion above regarding this point.

Additional comments

This version of the manuscript is greatly improved, with the addition of necessary control samples to the Figures, detailed explanation of the now appropriate statistics, better terminology and referencing of the literature, and the acknowledgment of study limitations with age as a confounding variable.

However, there remain several places in the manuscript that require improvements in clarity, the authors continue to overstate findings, and there are a number of other issues.

Some comments appear under BOTH Experimental Design and Validity of the Findings sections, because these were issues related to experimental design that influenced the validity of the authors' stated findings.

---

## Round 0.3 · accepted · Accept

Thank you for your consideration of the additional reviewer comments and your revised submission. I am happy to now accept your manuscript for publication in PeerJ. I have a few suggestions for changes that you could make when submitting the draft during production.

1. Thank you for supplying a supplemental figure comparing retinal sections from light exposed 9 and 18 month zebrafish. Unless I missed it I don’t see this figure referenced in the text of the manuscript. Please add a reference where it seems most appropriate. It might work best in the paragraph starting on line 420 that provides the rationale for the ages used in this study.

2. Line 322: It is unclear to me what you mean by “apparent resolution”. Is there a better word choice than “resolution”?

3. Line 336: Do you need to specify that you are referring to “red-green double cones” instead of just “cones”?

4. Lines 375: “immunolocalization” is misspelled.

5. Lines 394: Should this be “induced” by light?

You will be given the option to make the reviews of your manuscript available to readers. Please consider doing so as this review record can be a great resource for readers of your paper and contributes to more transparent science.

Thank you again for your contribution.

#